# DEEP INNOVATION PROTECTION

## ABSTRACT

Evolutionary-based optimization approaches have recently shown promising results in domains such as Atari and robot locomotion but less so in solving 3D tasks directly from pixels. This paper presents a method called *Deep Innovation Protection* (DIP) that allows training complex world models end-to-end for such 3D environments. The main idea behind the approach is to employ multiobjective optimization to temporally reduce the selection pressure on specific components in a world model, allowing other components to adapt. We investigate the emergent representations of these evolved networks, which learn a model of the world without the need for a specific forward-prediction loss.

## 1 INTRODUCTION

The ability of the brain to model the world arose from the process of evolution. It evolved because it helped organisms to survive and strive in their particular environments and not because such forward prediction was explicitly optimized for. In contrast to the emergent neural representations in nature, current world model approaches are often directly rewarded for their ability to predict future states of the environment (Schmidhuber, 1990; Ha & Schmidhuber, 2018; Hafner et al., 2018; Wayne et al., 2018). While it is undoubtedly useful to be able to explicitly encourage a model to predict what will happen next, in this paper we are interested in what type of representations can emerge from the less directed process of artificial evolution and what ingredients might be necessary to encourage the emergence of such predictive abilities.

In particular, we are building on the recently introduced world model architecture introduced by Ha & Schmidhuber (2018). This agent model contains three different components: **(1)** a visual module, mapping high-dimensional inputs to a lower-dimensional representative code, **(2)** an LSTM-based memory component, and **(3)** a controller component that takes input from the visual and memory module to determine the agent's next action. In the original approach, each component of the world model was trained separately and to perform a different and specialised function, such as predicting the future. While Risi & Stanley (2019) demonstrated that these models can also be trained end-to-end through a population-based genetic algorithm (GA) that exclusively optimizes for final performance, the approach was only applied to the simpler 2D car racing domain and it is an open question how such an approach will scale to the more complex 3D VizDoom task that first validated the effectiveness of the world model approach.

Here we show that a simple genetic algorithm fails to find a solution to solving the VizDoom task and ask the question what are the missing ingredients necessary to encourage the evolution of more powerful world models? The main insight in this paper is that we can view the optimization of a heterogeneous neural network (such as world models) as a *co-evolving system of multiple different sub-systems*. The other important insight is that representational innovations discovered in one subsystem (e.g. the visual system learns to track moving objects) require the other sub-systems to adapt. In fact, if the other systems are not given time to adapt, such innovation will likely initially have an adversarial effect on overall performance!

In order to optimize such co-evolving heterogeneous neural systems, we propose to reduce the selection pressure on individuals whose visual or memory system was recently changed, given the controller component time to readapt. This *Deep Innovation Protection* (DIP) approach is inspired by the recently introduced morphological innovation protection method of Cheney et al. (2018), which allows for the scalable co-optimization of controllers and robot body plans. Our approach is able to find a solution to the VizDoom: Take Cover task, which was first solved by the original

world model approach (Ha & Schmidhuber, 2018). More interestingly, the emergent world models learned to predict events important for the survival of the agent, even though they were not explicitly trained to predict the future. Additionally, our investigates into the training process show that DIP allows evolution to carefully orchestrate the training of the components in these heterogeneous architectures. We hope this work inspires more research that focuses on investigating representations emerging from approaches that do not necessarily only rely on gradient-based optimization.

## 2 DEEP INNOVATION PROTECTION

The hypothesis in this paper is that to optimize complex world models end-to-end for more complex tasks requires each of its components to be carefully tuned to work well together. For example, an innovation in the visual or memory component could adversely impact the controller component, leading to reduced performance. However, in the long run such innovation could allow an individual to outperform its predecessors.

The agent model is based on the world model approach introduced by Ha & Schmidhuber (2018). The network includes a sensory component, implemented as a variational autoencoder (VAE) that compresses the high-dimensional sensory information into a smaller 32-dimensional representative code (Fig. 1). This code is fed into a memory component based on a recurrent LSTM (Hochreiter & Schmidhuber, 1997), which should predict future representative codes based on previous information. Both the output from the sensory component and the memory component are then fed into a controller that decides on the action the agent should take at each time step. Following Risi & Stanley (2019), we train these world models end-to-end with a genetic algorithm, in which mutations add Gaussian noise to the parameter vectors of the networks: $\theta' = \theta + \sigma\epsilon$, where $\epsilon \sim N(0, I)$.

The approach introduced in this paper aims to train heterogeneous neural systems end-to-end by temporally reducing the selection pressure on individuals with recently changed modules, allowing other components to adapt. For example, in case of the world model, in which a mutation can either affect the VAE, MDN-RNN or controller, selection pressure should be reduced if a mutation affects the VAE or MDN-RNN, giving the controller time to readapt to the changes in the learned representation. Inspired by the multi-objective morphological innovation protection introduced by Cheney et al. (2018), we employ the well-known multiobjective optimization approach NSGA-II (Deb et al., 2002), in which a second "age" objective keeps track of when a mutation changes either the VAE or the MDN-RNN. Every generation an individual's age is increased by 1, however, if a mutation changes the VAE or MDN-RNN, this age objective is set to zero (lower is better). Therefore, if two neural networks reach the same performance (i.e. the same final reward), the one that had less time to adapt (i.e. whose age is lower) would have a higher chance of being selected for the next generation. The second objective is the accumulated reward received during an episode. Pseudocode of the approach applied to world models is shown in Algorithm 1.

It is important to note that this approach is different to the traditional usage of "age" in multi-objective optimization, in which age is used to increase diversity and keeps track of how long individuals have been in the population (Hornby, 2006; Schmidt & Lipson, 2011). In the approach in this paper, age counts how many generations the controller component of an individual had time to adapt to an unchanged visual and memory system.

In the original world model approach the visual and memory component were trained separately and through unsupervised learning based on data from random rollouts. In this paper they are optimized through a genetic algorithm without evaluating each component individually. In other words, the VAE is not directly optimized to reconstruct the original input data and neither is the memory component optimized to predict the next time step; the whole network is trained in an end-to-end fashion. Here we are interested in what type of neural representations emerge by themselves that allow the agent to solve the given task.

## 3 EXPERIMENTS

Following the original world model approach (Ha & Schmidhuber, 2018), in the experiments presented here an agent is trained to solve the car racing tasks, and the more challenging VizDoom task (Kempka et al., 2016) from 64×64 RGB pixel inputs (Fig. 2). In the continuous control task

---

**Algorithm 1** Deep Innovation Protection for World Models

---

1:  Generate random population of size N with age objectives set to 0
2:  **for** $generation = 1$ to $i$ **do**
3:      **for** Individual in Population **do**
4:          Objective[1] = age
5:          Objective[2] = accumulated task reward
6:          Increase individual's age by 1
7:      **end for**
8:      Assign ranks based on Pareto fronts
9:      Generate set of non-dominated solutions
10:     Add solutions, starting from first front, until number solution = N
11:     Generate child population through binary tournament selection and mutations
12:     Reset age to 0 for all individuals whose VAE or MDN-RNN was mutated
13: **end for**

---

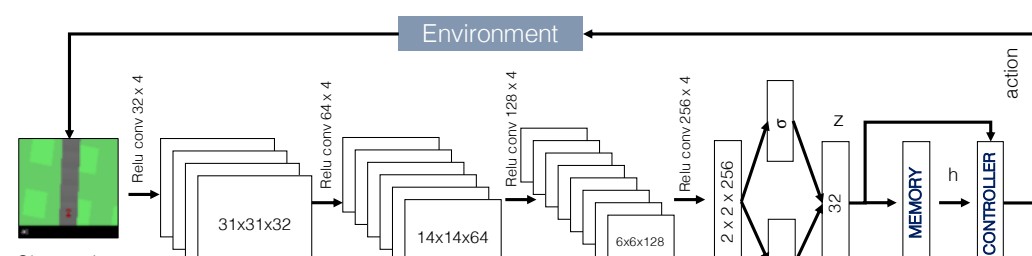

Figure 1: *Agent Model.* The agent model consists of three modules. A visual component (the encoder of a variational autoencoder) produces a latent code $z_t$ at each time step $t$, which is concatenated with the hidden state $h_t$ of the LSTM-based memory component that takes $z_t$ and previously performed action $a_{t-1}$ as input. The combined vector $(z_t, h_t)$ is input into the controller component to determine the next action of the agent. In this paper, the agent model is trained end-to-end with a multiobjective genetic algorithm.

`CarRacing-v0` (Klimov, 2016) the agent is presented with a new procedurally generated track every episode, receiving a reward of -0.1 every frame and a reward of $+100/N$ for each visited track tile, where $N$ is the total number of tiles in the track. The network controlling the agent (Fig. 1) has three outputs to control left/right steering, acceleration and braking. Further details on the network model, which is the same for both domains, can be found in the Appendix. In the `VizDoom:Take Cover` task the agent has to try to stay alive for 2,100 timesteps, while avoiding fireballs shot at it by strafing to the left or the right. The agent receives a +1 reward for every frame it is alive. The network controlling the agent has one output $a$ to control left ($a < -0.3$) and right strafing ($a > 0.3$), or otherwise standing still. In this domain, a solution is defined as surviving for over 750 timesteps, averaged across 100 random rollouts (Kempka et al., 2016).

Following the NSGA-II approach, individuals for the next generation are determined stochastically through 2-way tournament selection from the 50% highest ranked individuals in the population (Algorithm 1). No crossover operation was employed. The population size was 200. Because of the randomness in this domain, we evaluate the top three individuals of each generation one additional time to get a better estimate of the true elite. We compare a total of four different approaches:

1. **Deep innovation protection (DIP):** The age objective is reset to zero when either the VAE or MDN-RNN is changed. The idea behind this approach is that the controller should get time to readapt if one of the components that precede it in the network change.

2. **Controller innovation protection:** Here the age objective is set to zero if the controller changes. This setting tests if protecting components upstream can be effective in optimizing heterogeneous neural models.

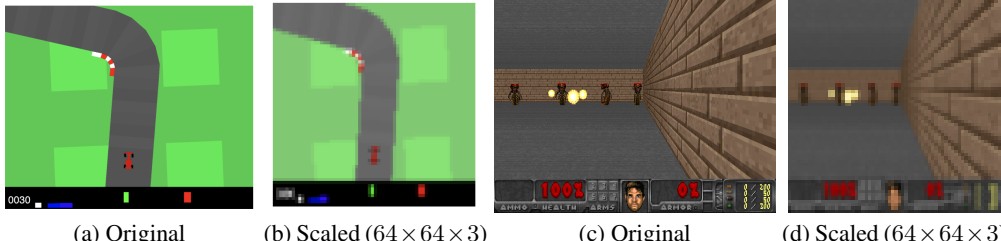

(a) Original      (b) Scaled ($64 \times 64 \times 3$)      (c) Original      (d) Scaled ($64 \times 64 \times 3$)

Figure 2: In the `CarRacing-v0` task the agent has to learn to drive across many procedurally generated tracks as fast as possible from $64 \times 64$ RGB color images. In the `VizDoom: Take Cover` domain the agent has to learn to avoid fireballs and to stay alive as long as possible.

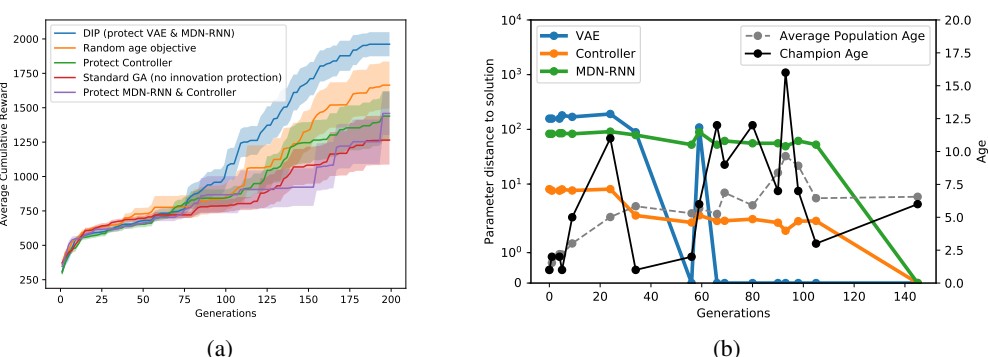

(a)                       (b)

Figure 3: *VizDoom Evolutionary Training.* Shown is (a) mean performance over generations together with one standard error. For a representative run of DIP (b), we plot the euclidean distances of the weights of the intermediate solutions (i.e. individuals with the highest task reward discovered so far) compared to the final solution in addition to their age and the average population age.

3. **MDN-RNN & Controller innovation protection:** This setup is the same as the controller protection approach but we additionally reset age if the MDN-RNN changes. On average, this treatment will reset the age objective as often as DIP.

4. **Random age objective:** In this setup the age objective is assigned a random number between [0, 20] at each evaluation. This treatment tests if better performance can be reached just through introducing more diversity in the population.

5. **Standard GA - no innovation protection:** In this non-multi-objective setup, which is the same one as introduced in Risi & Stanley (2019)[1], only the accumulated reward is taken into account when evaluating individuals.

For all treatments, a mutation has an equal probability to either mutate the visual, memory, or controller component of the network. Interestingly, while Risi & Stanley (2019) reported that this approach performs similarly well to an approach that always mutates all components, we noticed that it performs significantly worse in the more complicated VizDoom domain. This result suggests that the more complex the tasks, the more important it is to be able to fine-tune individual components in the overall world model architecture.

## 4 EXPERIMENTAL RESULTS

All results are averaged over ten independent evolutionary runs. In the car racing domain we find that there is no noticeable difference between an approach with and without innovation protection

---

[1]We use the publicly avaible code from Risi & Stanley (2019) for our baseline (https://github.com/uber-research/ga-world-models/).

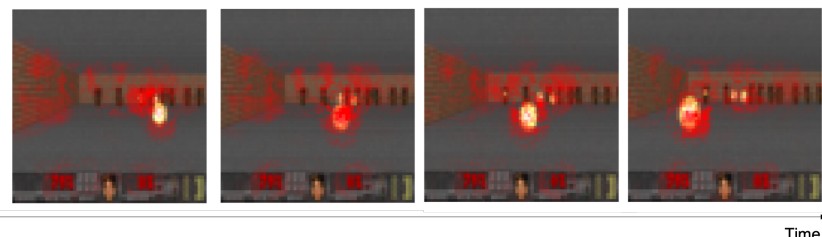

Time

Figure 4: *Still frames of a learned policy.* The evolved champion learned to primarily pay attention to the walls and fireballs, while ignoring the floor and ceiling. Interestingly the agent also seems to pay attention to the health and ammo indicator.

and both can solve the domain. However, in the more complex VizDoom task, the DIP approach that protects innovations in both VAE and MDN-RNN, outperforms all other approaches (Fig. 3a). The approach is able to find a solution to the task, effectively avoiding fireballs and reaching an average score of 824.33 (sd=491.59). To better understand the decision-making process of the agent, we calculate perturbation-based saliency maps (see Appendix for details) to determine the parts of the environment the agent is paying most attention to (Fig. 4). The idea behind perturbation-based saliency maps is to measure to what extent the output of the model changes if parts of the input image are altered (Greydanus et al., 2017). Not surprisingly, the agent learned to pay particular attention to the walls, fireballs, and the position of the monsters.

The better performance of the random age objective compared to no innovation protection suggests that increasing diversity in the population improves performance but less effectively than selectivity resetting the age objective as in DIP. Interestingly, the controller and the MDN-RNN&Controller protection approach perform less well, confirming our hypothesis that it is important to protect innovations upstream in the network for downstream components.

**Learned Representations**

We further investigate what type of world model can emerge from an evolutionary process that does not directly optimize for forward prediction or reconstruction loss. To gain insights into the learned representations we employ the t-SNE dimensionality reduction technique (Maaten & Hinton, 2008), which has proven valuable for visualizing the inner workings of deep neural networks (Such et al., 2018; Mnih et al., 2015). We are particularly interested in the information contained in the compressed 32-dimensional vector of the VAE and the information stored in the hidden states of the MDN-RNN (which are both fed into the controller that decides on the agent's action). Different combinations of sequences of these latent vectors collected during one rollout are visualized in two dimensions in Fig. 5. Interestingly, while the 32-dimensional $z$ vector from the VAE does not contain enough information to infer the correct action, either the hidden state alone or in combination with $z$ results in grouping the states into two distinct classes (one for moving left and one for moving right). The temporal dimension captured by the recurrent network proves invaluable in deciding what action is best. For example, not getting stuck in a position that makes avoiding incoming fireballs impossible, seems to require a level of forward prediction by the agent. To gain a deeper understanding of this issue we look more closely into the learned temporal representation next.

**Learned Forward Model Dynamics** In order to analyze the learned temporal dynamics of the forward model, we are taking a closer look at the average activation $x_t$ of all 256 hidden nodes at time step $t$ and how much they differ from the overall average across all time steps $\bar{X} = \frac{1}{N}\sum_1^N \bar{x}_t$. The variance of $\bar{x}_t$ is thus calculated as $\sigma_t = (\bar{X} - \bar{x}_t)^2$, and normalized to the range $[0, 1]$ before plotting. The hypothesis is that activation levels far from the mean might indicate a higher importance and should have a greater impact on the agent's controller component. In other words, they likely indicate critical situations in which the agent needs to pay particular attention to the predictions of the MDN-RNN. Fig. 6 depicts frames from the learned policies in two different situations, which shows that the magnitude of LSTM activations are closely tied to specific situations. The forward model does not seem to react to fireballs by themselves but instead depends on the agent being in the line of impact of an approaching fireball, which is critical information for the agent to stay alive.

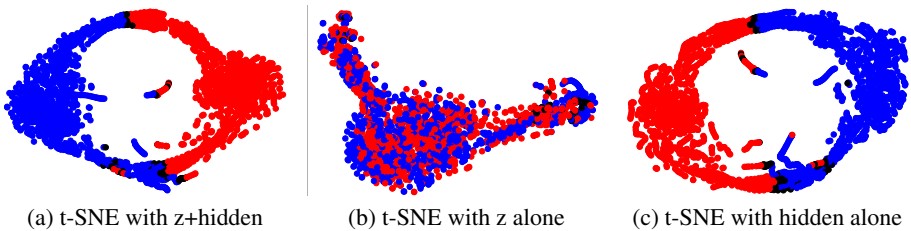

(a) t-SNE with z+hidden       (b) t-SNE with z alone       (c) t-SNE with hidden alone

Figure 5: t-SNE mapping of the latent+hidden vector (a), latent vector alone (b), and hidden vector alone (c). While the compressed latent vector is not enough to infer the correct action (b), the hidden LSTM vector alone contains enough information for the agent to decide on the correct action (c). Color legend: red = strafe left, blue = strafe right, black = no movement.

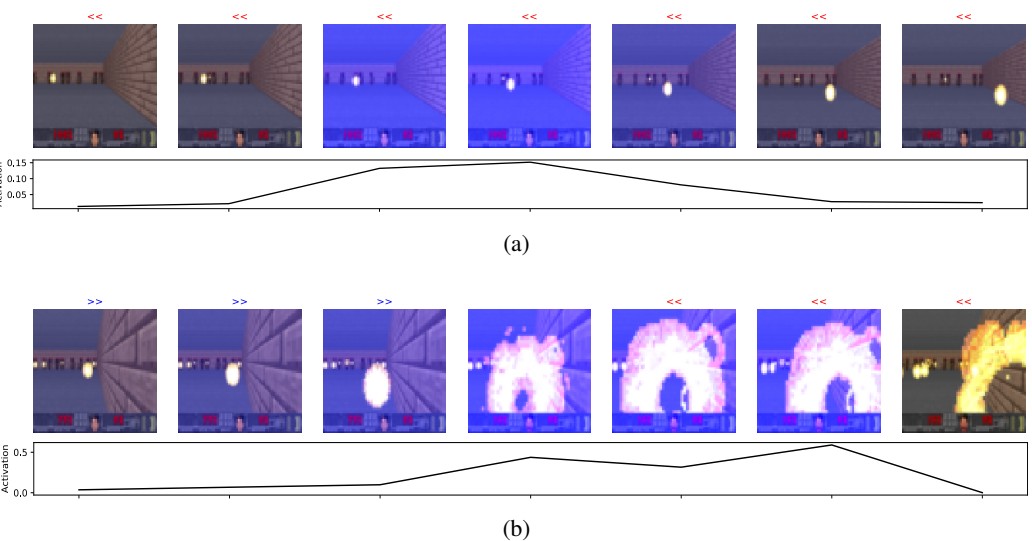

Figure 6: *Average activation levels of LSTM in two different situations.* For visualization purposes only, game images are colored more or less blue depending on the LSTM activation values. The evolved forward model seems to have learned to predict if a fireball would hit the agent at the current position. In (a) the agent can take advantage of that information to avoid the fireball while the agent does not have enough time to get out of the way in situation (b) and gets hit. Shown on top are the actions the agent takes in each frame.

**Evolutionary Innovations** In addition to analyzing the learned representations of the final networks, it is interesting to study the different stepping stones evolution discovered to solve the Viz-Doom task. We show one particular evolutionary run in Fig. 7, with other ones following similar progressions. In the first 30 generations the agent starts to learn to pay attention to fireballs but only tries avoiding them by either standing still or moving to the right. A jump in performance happens around generation 34 when the agent starts to discover moving to either the left or right; however, the learned representation between moving left or right is not well defined yet. This changes around generation 56, leading to another jump in fitness and some generations of quick fine-tuning later the agent is able to differentiate well between situations requiring different actions, managing to survive for the whole length of the episode. Motivated by the approach of Raghu et al. (2017) to analyse the gradient descent-based training of neural networks, we investigate the weight distances of the world model components of the best-performing networks found during training to the final solution representation (Fig. 3b). The VAE is the component with the steepest decrease in distance with a noticeable jump around generation 60 due to another lineage taking over. The MDN-RNN is optimized slowest, which is likely due to the fact that the correct forward model dynamics are more complicated to discover than the visual component. These results suggest that DIP is able to orchestrate the training of these heterogeneous world model architectures in an automated way.

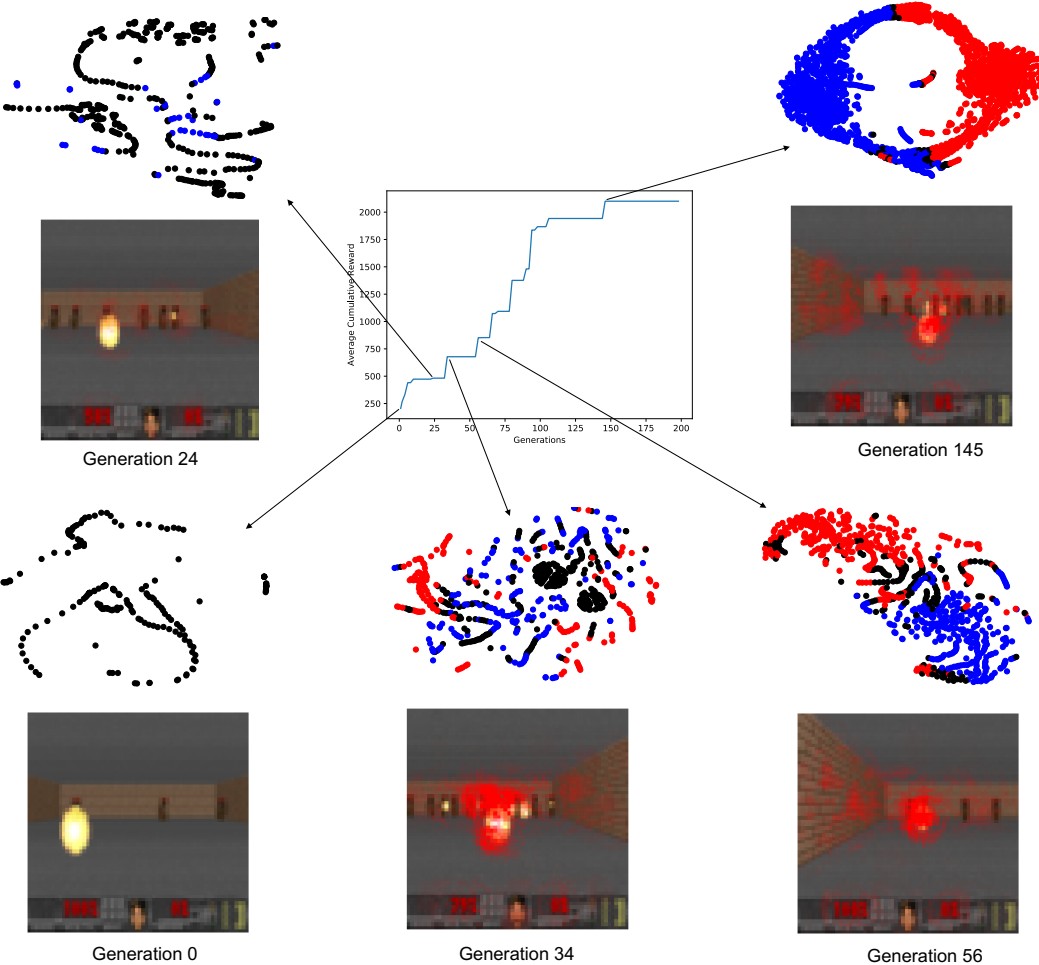

Figure 7: *Development of the evolved representation over generations.* Shown are t-SNE mappings of the 288-dimensional vectors (32-dimensional latent vectors + 256-dimensional hidden state vector) together with saliency maps of specific game situations. Early on in evolution the agent starts paying attention to the fireballs (generation 24) but only moves to the right (blue) or stands still (black). Starting around generation 34 the agent starts to move to the left and right, with the saliency maps becoming more pronounced. From generation 56 on the compressed learned representation (latent vector+hidden state vector) allows the agent to infer the correct action almost all the time. The champion discovered in generation 145 discovered a visual encoder and LSTM mapping that shows a clear division for left and right strafing actions.

## 5 RELATED WORK

A variety of different RL algorithms have recently been shown to work well on a diverse set of problems when combined with the representative power of deep neural networks (Mnih et al., 2015; Schulman et al., 2015; 2017). While most approaches are based on variations of Q-learning Mnih et al. (2015) or policy gradient methods (Schulman et al., 2015; 2017), recently evolutionary-based methods have emerged as a promising alternative for some domains (Such et al., 2017; Salimans et al., 2017). Salimans et al. Salimans et al. (2017) showed that a type of evolution strategy (ES) can reach competitive performance in the Atari benchmark and at controlling robots in MuJoCo. Additionally, Such et al. Such et al. (2017) demonstrated that a simple genetic algorithm is in fact able to reach similar performance to deep RL methods such as DQN or A3C. Earlier approaches that evolved neural networks for RL tasks worked well in complex RL tasks with lower-dimensional input spaces (Stanley & Miikkulainen, 2002; Floreano et al., 2008; Risi & Togelius, 2017).

However, when trained end-to-end these networks are often still orders of magnitude simpler than networks employed for supervised learning problems (Justesen et al., 2019) or depend on additional losses that are responsible for training certain parts of the network (Wayne et al., 2018).

For complex agent models, different network components can be trained separately (Wahlström et al., 2015; Ha & Schmidhuber, 2018). For example, in the world model approach (Ha & Schmidhuber, 2018), the authors first train a variational autoencoder (VAE) on 10,000 rollouts from a random policy to compress the high-dimensional sensory data and then train a recurrent network to predict the next latent code. Only after this process is a smaller controller network trained to perform the actual task, taking information from both the VAE and recurrent network as input to determine the action the agent should perform.

Evolutionary approaches solving 3D tasks directly from pixels has so far proven difficult although a few notable approaches exist. Koutník et al. (2013) evolved an indirectly encoded and recurrent controller for car driving in TORCS, which learned to drive based on a raw $64 \times 64$ pixel image. The approach was based on an indirect encoding of the network's weights analogous to the JPEG compression in images. To scale to 3D FPS tasks, Alvernaz & Togelius (2017) first trained an autoencoder in an unsupervised way and then evolved the controller giving the compressed representation as input. In another approach, Poulsen et al. (2017) trained an object recognizer in a supervised way and then in a separate step evolved a controller module. More recently, Lehman et al. (2018) introduced an approach called *safe mutations*, in which the magnitude of mutations to weight connections is scaled based on the sensitivity of the network's output to that weight. The safe mutations approach allowed the evolution of large-scale deep networks for a simple 3D maze task and is a complementary approach that could be combined with DIP in the future.

The approach introduced in this paper can be viewed as a form of diversity maintenance, in which selection pressure on certain mutated neural networks is reduced. Many other methods for encouraging diversity (Mouret & Doncieux, 2012) were invented by the evolutionary computation community, such as novelty search (Lehman & Stanley, 2008), quality diversity (Pugh et al., 2016), or speciation (Stanley & Miikkulainen, 2002). Using the concept of age to maintain diversity has a long history in evolutionary algorithms. Kubota & Fukuda (1997) first introduced the idea of using age in a genetic algorithm in which individuals are removed from the population if they reach a specific age limit. In the Age-Layered Population Structure approach introduced by Hornby (2006), the population is segregated into different age layers and newly generated individuals are introduced into the youngest age layer to increase diversity. Schmidt & Lipson (2011) combine the idea of age with a multi-objective approach, in which individuals are rewarded for performance and low age. Inspired by the morphological innovation approach (Cheney et al., 2018) and in contrast to previous approaches (Hornby, 2006; Schmidt & Lipson, 2011), DIP does not introduce new random individuals into the generation but instead resets the age of individuals whose sensory or memory system have been mutated. That is, it is not a measure of how long the individual has been in the population, as in the traditional usage of age in multi-objective optimization.

Approaches to learning dynamical models have mainly focused on gradient descent-based methods, with early work on RNNs in the 1990s (Schmidhuber, 1990). More recent work includes PILCO (Deisenroth & Rasmussen, 2011), which is a probabilistic model-based policy search method and Black-DROPS (Chatzilygeroudis et al., 2017), which employs CMA-ES for data-efficient optimization of complex control problems. Additionally, interest has increased in learning dynamical models directly from high-dimensional pixel images for robotic tasks (Watter et al., 2015; Hafner et al., 2018) and also video games (Guzdial et al., 2017). Work on evolving forward models has mainly focused on neural networks that contain orders of magnitude fewer connections and lower-dimensional feature vectors (Norouzzadeh & Clune, 2016) than the models in this paper.

## 6 DISCUSSION AND FUTURE WORK

The paper demonstrated that a world model representation for a 3D task can emerge under the right circumstances without being explicitly rewarded for it. To encourage this emergence, we introduced *deep innovation protection*, an approach that can dynamically reduce the selection pressure for different components in a heterogeneous neural architecture. The main insight is that when components upstream in the neural network change, such as the visual or memory system in a world model, components downstream need time to adapt to changes in those learned representations.

The neural model learned to represent situations that require similar actions with similar latent and hidden codes (Fig. 5 and 7). Additionally, without a specific forward-prediction loss, the agent learned to predict "useful" events that are necessary for its survival (e.g. predicting when the agent is in the line-of-fire of a fireball). In the future it will be interesting to compare the differences and similarities of emergent representations and learning dynamics resulting from evolutionary and gradient descent-based optimization approaches (Raghu et al., 2017).

Interestingly, without the need for a variety of specialized learning methods employed in the original world model paper, a simple genetic algorithm augmented with DIP can not only solve the simpler 2D car racing domain (Risi & Stanley, 2019), but also more complex 3D domains such as VizDoom. That the average score across 100 random rollouts is lower when compared to the one reported in the original world model paper (824 compared to 1092) is maybe not surprising; if random rollouts are available, training each component separately can results in a higher performance. However, in more complicated domains, in which random rollouts might not be able to provide all relevant experiences (e.g. a random policy might never reach a certain level), the proposed DIP approach could become increasingly relevant. An exciting future direction is to combine the end-to-end training regimen of DIP with the ability of training inside the world model itself (Ha & Schmidhuber, 2018). However, because the evolved representation is not directly optimized to predict the next time step and only learns to predict future events that are useful for the agent's survival, it is an interesting open question how such a different version of a hallucinate environment could be used for training.

A natural extension to this work is to evolve the neural architectures in addition to the weights of the network. Searching for neural architectures in RL has previously only been applied to smaller networks (Risi & Stanley, 2012; Stanley & Miikkulainen, 2002; Stanley et al., 2019; Gaier & Ha, 2019; Risi & Togelius, 2017; Floreano et al., 2008) but could potentially now be scaled to more complex tasks. While our innovation protection approach is based on evolution, ideas presented here could also be incorporated in gradient descent-based approaches that optimize neural systems with multiple interacting components end-to-end.

## A   APPENDIX

### A.1   OPTIMIZATION AND MODEL DETAILS

The size of each population was 200 and evolutionary runs had a termination criterion of 200 generations. The genetic algorithm $\sigma$ was determined empirically and set to 0.03 for the experiments in this paper. The code for the DIP approach can be found at: [Removed for anonymous review].

Table 1: *Number of parameters and training procedures.* The visual component of the agent (see Fig. 1) is effectively only utilizing and evolving the encoder part of the VAE, which has 755,744 parameters. The decoder network is composed of four deconvolutional layers and has 3,592,803 parameters.

| Model | #Params | WM Training Ha & Schmidhuber (2018) | GA Training |
|---|---|---|---|
| VAE | 4,348,547 | SGD - 1 epoch | |
| MDN-RNN | 384,071 | SGD - 20 epochs | Pop size 200 |
| Controller | 867 | CMA-ES - Pop 64 Rollouts 16 | Rollouts 1 |

An overview of the agent model is shown in Fig. 1, which employs the same architecture as the original world model approach Ha & Schmidhuber (2018). The sensory model is implemented as a variational autoencoder that compresses the high-dimensional input to a latent vector $z$. The VAE takes as input an RGB image of size $64 \times 64 \times 3$, which is passed through four convolutional layers, all with stride 2. Details on the encoder are depicted in the visual component shown in Fig. 1, where layer details are shown as: activation type (e.g. ReLU), number of output channels $\times$ filter size. The decoder, which is in effect only used to analyze the evolved visual representation, takes as input a tensor of size $1 \times 1 \times 104$ and processes it through four deconvolutional layers each with stride 2 and sizes of $128 \times 5$, $64 \times 5$, $32 \times 6$, and $32 \times 6$. The network's weights are set using the default

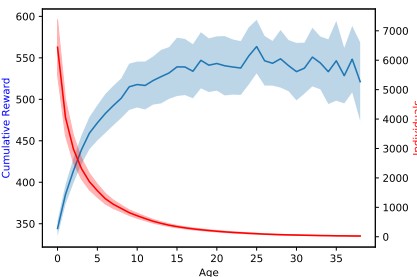

Figure 8: Average reward across ages and number of individuals per age.

PyTorch initilisation (He initialisation He et al. (2015)), with the resulting tensor being sampled from $\mathcal{U}(-\text{bound}, \text{bound})$, where bound $= \sqrt{\frac{1}{\text{fan\_in}}}$.

The memory model (Ha & Schmidhuber, 2018) combines a recurrent LSTM network with a mixture density Gaussian model as network outputs, known as a MDN-RNN (Ha & Eck, 2017; Graves, 2013b). The network has 256 hidden nodes and models $P(z_{t+1}|a_t, z_t, h_t)$, where $a_t$ is the action taken by the agent at time $t$ and $h_t$ is the hidden state of the recurrent network. Similar models have previously been used for generating sequences of sketches (Ha & Eck, 2017) and handwriting (Graves, 2013a). The controller component is a simple linear model that directly maps $z_t$ and $h_t$ to actions: $a_t = W_c[z_t h_t] + b_c$, where $W_c$ and $b_c$ are weight matrix and bias vector. Table 1 summarizes the parameter counts of the different world model components and how they are trained here and in the world model paper. The code for all the experiments is available at: [removed for anonymous review].

### A.2 INTERACTION BETWEEN REWARD AND AGE OBJECTIVE

We performed an analysis of the (1) cumulative reward per age and (2) the number of individuals with a certain age averaged across all ten runs and all generations (Fig. 8). While the average reward increases with age, there are fewer and fewer individuals at higher age levels. This result suggest that the two objectives (minimising age and increasing cumulative reward) are in competition with each other, motivating the choice for a multi-objective optimization approach. These results indicate that the multi-objective optimization is working as intended; staying alive for longer becomes increasingly difficult and a high age needs to be compensated for by a high task reward.

### A.3 SALIENCY MAP CALCULATION

Similarly to the approach by Greydanus et al. (2017), we calculate perturbation-based saliency maps by applying a Gaussian blur of $5 \times 5$ pixels to the coordinates $(i, j)$ of an image $I$ from the game. The Gaussian blur can be interpreted as adding uncertainty to a particular location of the screen. For example, if a fireball is at location $(i, j)$ then adding noise to that location makes the agent less certain about the fireball's location. The saliency map intensity $S(i, j)$ is calculated as the difference between the policy output $\pi$ given the original image $I$ and modified image $I'$ with added Gaussian blur at location $(i, j)$: $S(i, j) = |\pi(I) - \pi(I')|$.

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
