# OpenReview forum: "Deep Innovation Protection"
_ICLR.cc/2020/Conference — Reject_

### Official Review · AnonReviewer1 · 2019-10-06
**Official Blind Review #1**

**Rating:** 6

**Review:**

This work improves upon the current state of applying neuroevolution to solve RL tasks, by using a modular "world models" architecture. While the original work used backpropagation to train the three different components on different tasks, later work used neuroevolution to train the entire architecture for final performance; however, the latter work was unable to demonstrate success on the more challenging task used in the original work. Given that the architecture has a heterogenous pipeline, the authors propose, and empirically show (over a set of seeds), that allowing extra time for components to adapt to other, more recently-changed components, enables neuroevolution to solve the second task. I would give this paper an accept, as this insight, along with a novel algorithm, could aid further work in neuroevolution but also more generally the optimisation of modular NNs.

The ablations are an important part of this work, and it would be good to study these further. The random age objective performing well does indeed indicate that diversity preservation plays a role in the better performance in DIP, but the fact that protecting the upstream controller does improve performance slightly makes me question if this effect has been properly decoupled. This ablation will reset the age 1/2 the time DIP does, reducing diversity/increasing selection pressure, and so another appropriate ablation is to see the effects of protecting MDN-RNN and controller innovations (with the VAE being the most downstream component).

**Experience Assessment:**

I have read many papers in this area.

**Review Assessment: Checking Correctness Of Derivations And Theory:**

N/A

**Review Assessment: Checking Correctness Of Experiments:**

I carefully checked the experiments.

**Review Assessment: Thoroughness In Paper Reading:**

I read the paper thoroughly.

---

> ### Author Response · Authors · 2019-11-12
> **Response to Review #1**
>
> We thank the reviewer for the encouraging comments and this very relevant additional experiment suggestion. We now ran the proposed experiment and added the results to the paper (Figure 3a). The MDN-RNN&Controller protection approach does perform significantly worse than DIP, confirming the hypothesis that protecting the most downstream controller component is crucial for the success of our approach.

---

### Official Review · AnonReviewer3 · 2019-10-24
**Official Blind Review #3**

**Rating:** 3

**Review:**

This paper is well organized. The applied methods are introduced in detail. But it lacks some more detailed analysis.

My concerns are as follows.
1. The idea of using the concept of age is not new. There are many studies that have been conducted using the concept of aging in the literature of evolutionary algorithms.
2. Multi-objective optimizations involve finding a set of optimal solutions for different and often competing objectives. However, the authors did not provide the analysis that the age and the accumulated task reward are conflicting with each other.
3. It would be better the average survival time of the individuals is presented in this paper.

**Experience Assessment:**

I have read many papers in this area.

**Review Assessment: Checking Correctness Of Derivations And Theory:**

I assessed the sensibility of the derivations and theory.

**Review Assessment: Checking Correctness Of Experiments:**

I assessed the sensibility of the experiments.

**Review Assessment: Thoroughness In Paper Reading:**

I read the paper at least twice and used my best judgement in assessing the paper.

---

> ### Author Response · Authors · 2019-11-11
> **Response to Review #3**
>
> Thank you very much for your comments. We agree with the assessment that a more detailed analysis was needed and added the requested information to the manuscript, as detailed below.
>
> Q1: We have now significantly extended the related work section to include articles on age in EAs, while also highlighting how our approach is significantly different to previous work in Section 2. In short, the important difference in our work is that age counts how many generations the controller component of an individual has had to adapt to an unchanged visual and memory system, not for how many generations an individual was in the population (the typical approach in age-based multi-objective approaches). Please also see the first paragraph of the response to Review #2 for a more detailed explanation.
>
> The related work section now reads: “Using the concept of age to maintain diversity has a long history in evolutionary algorithms. Kubota & Fukuda (1997) first introduced the idea of using age in a genetic algorithm in which individuals are removed from the population if they reach a specific age limit. In the Age-Layered Population Structure approach introduced by Hornby (2006), the population is segregated into different age layers and newly generated individuals are introduced into the youngest age layer to increase diversity. Schmidt & Lipson (2011) combine the idea of age with a multi-objective approach, in which individuals are rewarded for performance and low age.  Inspired by the morphological innovation approach (Cheney et al., 2018) and in contrast to previous approaches (Hornby, 2006; Schmidt & Lipson, 2011), DIP does not introduce new random individuals into the generation but instead resets the age of individuals whose sensory or memory system have been mutated. That is, it is not a measure of how long the individual has been in the population, as in the traditional usage of age in multi-objective optimization.”
>
> Q2 and Q3: We have now added a more detailed analysis of how the age and reward objectives interact and how this affects the agent’s survival time (Section A.2 in the appendix). In Figure 8 we show that both objectives are indeed in competition with each other, further motivating the use of a multi-objective optimization approach. While the average reward increases with age, there are fewer and fewer individuals at higher age levels. These results indicate that the multi-objective optimization is working as intended; staying alive for longer becomes increasingly difficult and a high age needs to be compensated for by a high task reward. We additionally now include the age of the best performing individual and the average population age in Figure 3b, which shows that it is indeed not always the oldest individual that has the highest fitness.

---

### Official Review · AnonReviewer2 · 2019-10-24
**Official Blind Review #2**

**Rating:** 6

**Review:**

Review of “Deep Innovation Protection”

This paper builds on top of the World Models line of work that explores in more detail the role of evolutionary computing (as opposed to latent-modeling and planning direction like in Hafner2018) for such model-based architectures. Risi2019 is the first work that is able to train the millions of weights of the architecture proposed in Ha2018 end-to-end using modular-form of genetic algorithm (GA) to maximize the terminal cumulative reward.

This work expands on Risi2019, and proposes the use of multi-objective optimization (NSGA-II) to allow a world model to learn several subsystems by the fact that the GA optimizer is not only targeting the reward function of the environment, but it is also optimizing for diversity, similar to MAP-Elites line of literature (i.e. Cully2015, Mouret2015, and other more recent cited in this paper).

They demonstrate that end-to-end multi-objective optimization is not only able to solve both CarRacing and VizDoom tasks, but the world models learned have interesting features, i.e. the emerged world models can predict important survival events even though they were not trained explicitly for forward prediction.

While the paper is interesting, I feel in its current state, it is not the level of an ICLR conference paper. Below is some feedback I want to give the authors to help them improve the work, so hopefully it can be improved either for this conference or the next. For the record, the score I wanted to give is a 4, but since I can only choose 3, or 6, I will assign a score of 3 for this review (although should really be a 4). I will breakdown the feedback into minor and major:

Minor feedback:

- Figure 3 does not indicate which task it is solving. Is it CarRacing or VizDoom?

- What's the motivation for optimizing for both "age" and reward? Why not other multi-objectives?

Medium Feedback:

As the authors are learning world models that can predict some form of a future, can they train agents inside an open-loop environment generated by such a world model (perhaps even with some hacks), and transfer such a policy back to the real env, as done in Ha2018 for VizDoom?

Major feedback:

The contribution of this paper isn't there, compared to Risi2019 which I feel has a much larger contribution to the line of work. I'm not convinced that the approach (multi-obj + end-to-end) which solves VizDoom cannot be solved using Risi2019. Authors mentioned this in the intro, but perhaps show some convincing experiments to make sure Risi2019 will definitely fail VizDoom? I'm not really convinced that an end-to-end algorithm will fail to find a solution for a poilcy (with our without a world model component in the architecture) for VizDoom Take Cover. Honestly I don't think VizDoom Take Cover is that hard of a task, and my intuition tells me that the method outlined in Risi2019, if carefully tuned properly, will be able to solve VizDoom Take Cover (but happy to be proven wrong if there is a strong experimental argument suggesting otherwise).

I think for this paper to be convincing, the authors actually need to go beyond the CarRacing and VizDoom tasks and show the real power of multi-objective optimization for solving difficult problems that require model-based architectures. For instance, in Cully2015, the learning of several diverse sub-systems on the Pareto front allowed a robot to still be able to accomplish the walking task when the legs are disabled. I would advise the authors to look for a third task where they can clearly show that their approach can solve problems that current approaches (Ha2018, Risi2019, and possibly even DeepRL) cannot solve. Being able to do this has the potential to make this paper into an important work for the field. I can also recommend trying out some environments in the Animal Olympics (see refs) that will be very much suited for multi-obj optimization for animal-like survival environments where input space are pixels.

[Ha2018] Recurrent World Models Facilitate Policy Evolution (NeurIPS2018)

[Risi2019] Deep neuroevolution of recurrent and discrete world models (GECCO2019)

[Hafner2018] Learning latent dynamics for planning from pixels (PlaNet)

[Cully2015] “Robots that can adapt like animals” https://www.nature.com/articles/nature14422

[Mouret2015] Illuminating search spaces by mapping elites

[Animal Olympics] http://animalaiolympics.com/

*** REVISED SCORE ***

Upon looking at the revision, and the author response, I decided to increase the score of the paper (for the record: originally from a 4, which is rounded to a 4, to a 5, which is rounded to a 6).

The authors explained why Risi2019's approach cannot work on VizDoom with sufficient effort, and I buy their explanation. At the same time I feel this work is can benefit from addition, convincing tasks that highlight the full power of multi objective optimization, for this work to warrant a strong accept. Regardless of the final decision, I'm looking forward to see development of this interesting approach.

**Experience Assessment:**

I have published one or two papers in this area.

**Review Assessment: Checking Correctness Of Derivations And Theory:**

I assessed the sensibility of the derivations and theory.

**Review Assessment: Checking Correctness Of Experiments:**

I assessed the sensibility of the experiments.

**Review Assessment: Thoroughness In Paper Reading:**

I read the paper at least twice and used my best judgement in assessing the paper.

---

> ### Author Response · Authors · 2019-11-11
> **Response to Review #2**
>
> Thank you very much for your comments. Regarding the minor and medium feedback: We have now changed the caption of Figure 3 to clarify that it shows performance on the VizDoom task. More importantly, we realize that our initial submission did not make clear enough the difference of our approach from the traditional usage of ‘age’ in multi-objective optimization. Typically, and as noted by the reviewer, “age” in multi-objective optimization is used to increase diversity and to keep track of how long individuals have been in the population (Hornby, 2006; Schmidt & Lipson, 2011). The important difference in our work is that age counts how many generations the controller component of an individual has had to adapt to an unchanged visual and memory system.  That is, it is not a measure of how long the individual has been in the population. Only when the visual or memory system changes is the age of an individual reset to 0. This way the controller is actually given time to adapt to representational innovations discovered in the memory or visual system (e.g. the visual system learns to track moving objects); if the controller would not be given this time to adapt, such innovations would likely initially have an adverse effect on overall performance even though they are beneficially in the long run. We have now clarified this uniqueness in Section 2, and we believe it is one of the main and novel contributions of this paper.
>
> As the reviewer notes, other objectives could potentially be added as well, such as Novelty, which could further improve performance. However, for this paper we first wanted to investigate potential improvements from adding this novel form of age alone, before we will explore more complicated setups in the future.
>
> Regarding training inside the world model, as was done in Ha2018, we believe this is an exciting direction we would like to explore more in the future but felt it was outside the scope of this paper. Because the evolved representation is not directly optimized to predict the next time step and only learns to predict future events that are useful for the agent’s survival, it is an interesting open question how such a different version of a hallucinate environment could be used for training. We added a discussion on this direction to Section 6 of the paper.
>
> In response to the major feedback, it might not have been clear from our initial submission that we indeed use the exact same non-multi-objective approach from Risi2019 to try to solve the VizDoom task (building on their publicly available Github code). These training results are shown in Figure 3 as “No innovation protection”. We tried a variety of different mutation rates but the results always looked similar to the one shown in Figure 3; the new approach presented in our paper performed significantly better to the approach from Risi2019 in the more complicated VizDoom task. In testing (100 rollouts), none of the 10 runs with the Risi2019 approach was able to solve the domain (reaching an average score of 750).
>
> Therefore, we would argue that the VizDoom task, which validated the original approach published by Ha at NeurIPS, is indeed sufficiently complex enough to show the usefulness of our new approach. Reaching a score of more than 750 across 100 random rollouts is in fact challenging, and as far as we are aware, so far only solved by Ha2018 and our approach.

---

### Decision · Program_Chairs · 2019-12-19

**Decision:**

Reject

**Comment:**

This paper is a very borderline case. Mixed reviews. R2 score originally 4, moved to 5 (rounded up to WA 6), but still borderline. R1 was 6 (WA) and R3 was 3 (WR).  R2 expert on this topic, R1 and R3 less so. AC has carefully read the reviews/rebuttal/comments and looked closely at the paper. AC feels that R2's review is spot on and that the contribution does not quite reach ICLR acceptance level, despite it being interesting work. So the AC feels the paper cannot be accepted at this time. But the work is definitely interesting -- the authors should improve their paper using R2's comments and resubmit.